

# Effects of knee extension with different speeds of movement on muscle and cerebral oxygenation

Damiano Formenti[1], David Perpetuini[2,3], Pierpaolo Iodice[4,5],
Daniela Cardone[2,3], Giovanni Michielon[1], Raffaele Scurati[1],
Giampietro Alberti[1] and Arcangelo Merla[2,3]

[1] Department of Biomedical Sciences for Health, Università degli Studi di Milano, Milan, Italy
[2] Department of Neuroscience, Imaging, and Clinical Sciences, University "G. d'Annunzio" Chieti-Pescara, Chieti, Italy
[3] Infrared Imaging Lab, Centro ITAB-Institute for Advanced Biomedical Technologies, University "G. d'Annunzio" Chieti-Pescara, Chieti, Italy
[4] Institute of Cognitive Sciences and Technologies, National Research Council, Rome, Italy
[5] Centre d'Etude des Transformations des Activités Physiques et Sportives (CETAPS), University of Rouen Normandy, Mont-Saint-Aignan, France

Corresponding author
Damiano Formenti,
damiano.formenti@unimi.it

## ABSTRACT

**Background.** One of the mechanisms responsible for enhancing muscular hypertrophy is the high metabolic stress associated with a reduced muscular oxygenation occurring during exercise, which can be achieved by reducing the speed of movement. Studies have tested that lowered muscle oxygenation artificially induced by an inflatable cuff, could provoke changes in prefrontal cortex oxygenation, hence, to central fatigue. It was hypothesized that (1) exercising with a slow speed of movement would result in greater increase in cerebral and greater decrease in muscle oxygenation compared with exercises of faster speed and (2) the amount of oxygenation increase in the ipsilateral prefrontal cortex would be lower than the contralateral one.

**Methods.** An ISS Imagent frequency domain near infrared spectroscopy (NIRS) system was used to quantify oxygenation changes in the vastus lateralis muscle and prefrontal cortex (contra- and ipsilateral) during unilateral resistance exercises with different speeds of movement to voluntary fatigue. After one maximal repetition (1RM) test, eight subjects performed three sets of unilateral knee extensions (∼50% of 1RM), separated by 2 min rest periods, following the pace of 1 s, 3 s and 5 s for both concentric and eccentric phases, in a random order, during separate sessions. The amount of change for NIRS parameters for muscle ($\Delta$Hb: deoxyhemoglobin, $\Delta$HbO: oxyhemoglobin, $\Delta$HbT: total hemoglobin, $\Delta$StO$_2$: oxygen saturation) were quantified and compared between conditions and sets by two-way ANOVA RM. Differences in NIRS parameters between contra- and ipsilateral (lobe) prefrontal cortex and conditions were tested.

**Results.** Exercising with slow speed of movement was associated to larger muscle deoxygenation than normal speed of movement, as revealed by significant interaction (set × condition) for $\Delta$Hb ($p = 0.01$), and by significant main effects of condition for $\Delta$HbO ($p = 0.007$) and $\Delta$StO$_2$ ($p = 0.016$). With regards to the prefrontal cortex, contralateral lobe showed larger oxygenation increase than the ipsilateral one for $\Delta$Hb, $\Delta$HbO, $\Delta$HbT, $\Delta$StO$_2$ in each set (main effect of lobe: $p < 0.05$). Main effects of

condition were significant only in set1 for all the parameters, and significant interaction lobe × condition was found only for $\Delta$Hb in set1 ($p < 0.05$).

**Discussion**. These findings provided evidence that speed of movement influences the amount of muscle oxygenation. Since the lack of oxygen in muscle is associated to increased metabolic stress, manipulating the speed of movement may be useful in planning resistance-training programs. Moreover, consistent oxygenation increases in both right and left prefrontal lobes were found, suggesting a complementary interaction between the ipsi- and contralateral prefrontal cortex, which also seems related to fatigue.

## INTRODUCTION

Maximizing hypertrophic response to resistance training can be reached by manipulation of exercise program variables, such as type and order of exercises, length of rest intervals, intensity of maximal load, and training volume (*Kraemer & Ratamess, 2004*). Recently, research attention has been devoted to repetition duration (i.e., the sum of the concentric, eccentric, and isometric actions of a repetition), a variable that has often been ignored (*Headley et al., 2011*). Substantially, current evidence suggests that hypertrophic outcomes are similar when training with repetition durations from 0.5 to 8.0 s to concentric failure (*Schoenfeld, Ogborn & Krieger, 2015*). However, the review by *Schoenfeld, Ogborn & Krieger (2015)* considered studies employing protocols with different intensities, which may have masked the role of repetition duration in muscle hypertrophy. Several studies have addressed the effects of low-intensity resistance exercise with slow movement (i.e., longer repetition duration) on muscular size and strength (*Tanimoto & Ishii, 2006*; *Tanimoto et al., 2008*; *Watanabe et al., 2013*). This modality of resistance training is characterized by relatively low intensity (∼50% of one maximal repetition (1RM)) with repetition durations of 7 s (3 s eccentric, 3 s concentric, and 1 s isometric contraction, no relaxation time). This has been shown to contribute to increased muscular hypertrophy and strength gains similar to conventional high intensity resistance training with normal speed (1–2 s for eccentric and concentric phase), even with relatively low intensity (∼50% of 1RM) (*Tanimoto & Ishii, 2006*; *Tanimoto et al., 2008*; *Watanabe et al., 2013*). In fact, one of the mechanisms responsible for enhancing muscular hypertrophy is the lowered peripheral muscular oxygenation occurring during exercise (*Tamaki et al., 1994*), which can be obtained naturally by reducing the speed of movement (*Tanimoto, Madarame & Ishii, 2005*; *Tanimoto & Ishii, 2006*). It has been proposed that muscle hypertrophy is likely attributed to mechanical tension and to the increased levels of metabolic stress within muscle (as a result of the intramuscular environment with reduced oxygenation) (*Pearson & Hussain, 2015*), which in turn stimulates growth hormone (GH) secretion (*Takarada et al., 2000*), cell swelling (*Loenneke et al., 2012*), production of reaction oxygen species (ROS) (*Kawada & Ishii, 2005*), and increased recruitment of fast-twitch muscle fibers

(*Moritani et al., 1992*). A much lower peripheral muscle oxygenation can be likely achieved by further lowering the speed of movement, thus increasing the repetition duration and maximizing the hypertrophic response. Furthermore, it has been suggested that a reduced muscular oxygenation elicits early recruitment of fast twitch-fibers because of early fatigue (*Husmann et al., 2017*). Moreover, a reduced muscular oxygenation during resistance exercise to fatigue has been found to be related with increase in deoxyhemoglobin concentration and lower increase in oxyhemoglobin concentration in prefrontal cortex, thus suggesting that the ability to sustain exercise to exhaustion is related to both muscle and cerebral oxygenation (*Ganesan et al., 2015*). In fact, a decrease in oxygenation of the prefrontal cortex at exhaustion may lead to the decision to stop exercising (*Shibuya & Tachi, 2006*; *Subudhi, Dimmen & Roach, 2007*; *Subudhi et al., 2009*).

Recent studies have examined the cerebral and muscle oxygenation changes during sustained isometric (*Pereira, Gomes & Bhambhani, 2009*) and dynamic knee extension to voluntary fatigue (*Matsuura et al., 2011*). It was found that fatigue during unilateral knee extensions performed under isometric and isotonic conditions was not related to a decline in neuronal activation but rather mediated peripherally in the exercising muscle (*Pereira, Gomes & Bhambhani, 2009*). In fact, cerebral oxygenation increased systematically during the exercise with no signs of attenuation. Hence, muscle deoxygenation may be directly responsible for the inability to sustain adequate muscle contractions at exhaustion and to inhibit central motor output via peripheral feedback system (*Gandevia, 2001*). On the other hand, since cerebral oxygenation reflects modulations in cerebral functional activation (*Colier et al., 1999*), a decreased cerebral oxygenation may affect maximal exercise capacity, which highlights the key role of cerebral oxygenation in modulating motor output at exhaustion (*Shibuya & Tachi, 2006*; *Subudhi, Dimmen & Roach, 2007*; *Subudhi et al., 2009*).

Furthermore, it has been demonstrated that during handgrip exercise the ipsilateral and contralateral frontal lobes have different oxygenation dynamics (*Kuboyama & Shibuya, 2015*) and blood flow (*Fernandes et al., 2016*). It is licit to suppose that unilateral resistance training could induce differences in contralateral and ipsilateral frontal cortical oxygenation.

Hitherto, to the best of our knowledge, there are no studies that have investigated whether resistance exercise with different repetition durations (and different speed of movement) would induce differences in muscle oxygenation, as well as differences in ipsilateral and contralateral prefrontal cortex oxygenation. We hypothesized that measurements of muscle hemodynamic and oxygenation during resistance exercises with different repetition durations with the same intensity would provide new insights into these processes. In this study we monitored muscle and prefrontal cortex oxygenation through Near Infrared Spectroscopy (NIRS) to investigate tissue oxygenation responses to resistance exercise to voluntary fatigue, performed with different speeds of movement.

Therefore, the purposes of the current study were: (1) to investigate the effect of resistance exercises with different speeds of movement (i.e., different repetition duration) on muscle and cerebral oxygenation; (2) to investigate the difference in oxygenation change

between the ipsilateral and contralateral prefrontal cortex during resistance exercises to voluntary fatigue.

We hypothesized that: (1) the exercise with slow speed of movement (longer repetition duration) would result in greater increase in cerebral oxygenation and greater decrease in muscle oxygenation compared with exercises of faster speed; (2) during the exercises, the amount of oxygenation increase in the ipsilateral prefrontal cortex would be lower compared to the contralateral one.

## MATERIAL AND METHODS

### Participants

Eight young healthy female subjects, who were not involved in regular resistance training programs at the time of the study, volunteered to participate. The subjects were sub-elite futsal players. The choice of a futsal team permitted to have high interindividual homogeneity of anthropometric characteristics and training conditions. Their age, body mass and stature were $24.8 \pm 4.9$ years, $63.3 \pm 7.2$ kg and $164 \pm 4.7$ cm (mean $\pm$ SD). None of the participants reported recent lower limb injuries. According to the declaration of Helsinki, the study was approved by the Ethical Committee of the Università degli Studi di Milano (approval number: 2/12). After a thorough explanation of the protocol, the subjects provided informed written consent to participate in this study.

The sample size was calculated based on a power analysis (G*Power 3.1.9.2 http://www.gpower.hlu.de/en.html) conducted on the effect size (Cohen's $f$; varying between 0.85 and 0.89) calculated by data reported in a previous study on the same topic (*Ganesan et al., 2015*). The analysis was based on the following parameters: $\eta_{p^2} = 0.43$; alpha $= 0.05$; power $= 0.95$; number of groups/measures: depends on the specific analysis (*Cohen, 1988*; *Richardson, 2011*). The results of these analyses indicated a required sample size ranging from seven to nine participants; we observed eight to be a critical number of participants, as the effect size and observed power remains stable above this number.

### Experimental protocol

We used a within-subjects design, so that each subject served as their own control. The experiment comprised four different sessions, each one separated by five days. After a preliminary session, the three experimental sessions, each one involving different conditions—denoted as 1 s exercise, 3 s exercise and 5 s exercise—were randomized. The subjects were instructed to refrain from strenuous physical activity in the two days before the trials and abstained from consuming alcoholic or caffeine-containing products for a 4 h period before the start of the experiment. All sessions were scheduled in the late morning to mitigate possible effects due to circadian rhythm variations. A schematic representation of the experimental design is shown in Fig. 1.

### Preliminary session

The purpose of the preliminary session was to collect anthropometric measurements, to find out the load of maximal repetition (1RM) in knee extension of the dominant leg, as well as to familiarize the subjects with different speeds of movement (i.e., 1 s, 3 s and 5 s

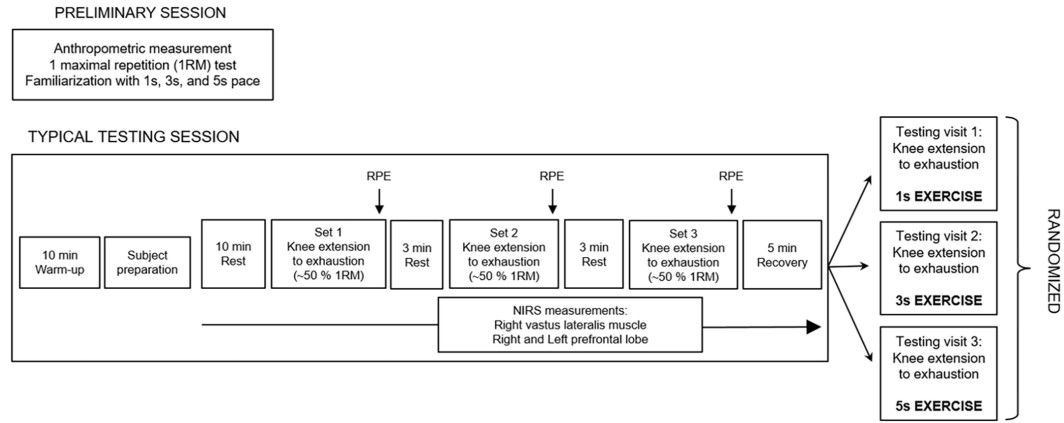

**Figure 1** **Schematic representation of the experimental design.** The preliminary session and each testing session were performed in separate days. 1 s exercise, 1 s concentric and 1 s eccentric phase. 3 s exercise, 3 s concentric and 3 s eccentric phase. 5 s exercise, 5 s concentric and 5 s eccentric phase.

exercises). All the subjects were right-leg dominant. Participants warmed up by completing a number of submaximal repetitions (~15) at approximately 20% of the perceived 1 RM. After 60 s of rest, an initial weight within the subject's perceived capacity (~50%–70%) was chosen. Resistance was progressively increased by 10.0%–20.0% from the previous successful attempt until the subjects could not complete the single repetition throughout the full range of motion. The range of motion was set to 105° (i.e., from starting position of 75° to ending position of 180°). The 1 RM was determined as the maximum weight that the subjects were able to lift once. A rest period of 300 s was given to participants between successive trials, and in all of the cases the 1 RM was obtained within five trials. After this assessment, the subjects were familiarized with the resistance exercise protocol used in the subsequent three testing sessions. Participants performed three sets of knee extensions, using ~30% of their 1 RM, with each set lasting approximately 60 s. Each set was performed using a specific speed of movement (1 s/1 s, 3 s/3 s, 5 s/5 s for concentric and eccentric condition respectively, randomly administrated) with the aid of a metronome.

## Testing visits: 1 s, 3 s and 5 s exercise sessions

Before the trials, subjects performed a standardized warm-up, i.e., 10 min of walking on a treadmill. After that, and before the beginning of the exercises, subjects acclimated to the room conditions (temperature 22–24 °C; relative humidity 50 ± 5%; no direct ventilation and constant intensity of light) for 10 min, at rest, remaining seated on the knee extension machine (Teca srl, Ortona, Italy). Then, subjects underwent either the 1 s, 3 s or 5 s knee extension exercises (the three conditions were randomized and separated by at least 5 days) using an intensity of ~50% of 1RM. Each exercise session consisted of three sets of knee extensions with an inter-set rest period of 3 min.

In either session, subjects repeated the movement until exhaustion (concentric failure) following the 1 s/3 s/5 s pace for each phase of contraction. A metronome was used to ensure that each repetition duration lasted 2 s, 6 s, and 10 s for each of the 1 s/3 s/5 s

conditions respectively. Shoulder straps were used to stabilize the subjects and to minimize the use of trunk muscles during the exercises. At the end of each exercise set, subjects were asked to provide a rating of perceived exertion (RPE), expressed as a number between 1 and 10 (*Day et al., 2004*).

## Near infrared spectroscopy measurements

Muscle and cerebral oxygenation were assessed using a frequency-domain near-infrared spectroscopy machine, with two wavelengths of near-infrared light (690 and 830 nm), at a sampling rate of five Hz (ISS Imagent, ISS, Champaign, IL, USA). A multi-distance optical probe, configured with one optical detector and four optical source fibers, was attached to the muscle. The fibers were positioned on a custom-made probe such that there are four source-detector separation distances (2.0, 2.5, 3.0, 3.5 cm) for each wavelength. The four sources were on the same line of the detector. The probe was positioned on the belly of the vastus lateralis of the right limb, approximately 12 cm from the lateral epicondyle of the femur, as used in previous resistance exercise studies (*Gomes, Matsuura & Bhambhani, 2012*; *Yeung et al., 2016*). To ensure the reproduction of the probe position in the following procedures, pen-marks were made around the margins of the probe. The probe was secured with tape and black bandages were wrapped around the leg to block background light. Skinfold thickness at the site of probe placement (on vastus lateralis) was measured by a skinfold caliper, in order to determine adipose tissue thickness, defined as skinfold thickness/2. The source-detector separation distances of the probes allowed to investigate up to 18 mm of depth. The adipose tissue thickness (7.2 ± 1.2 mm) was lower than the NIRS light penetration depth, and therefore NIRS measurements were not influenced by adipose tissue (*Van Beekvelt et al., 2001*).

In order to record the oxygenation of both the contralateral and ipsilateral prefrontal cortex (*Chiarelli et al., 2016*), a probe composed of two detectors and eight sources with the same source-detector distance configuration of the muscle probe was used. The center of the probe was positioned in correspondence with the *Fp* point (fronto polar), which is a marker defined in the 10–20 international system for the placement of the Electroencephalography (EEG) probes. It is placed proximally to the nasion (root of nose), in a position equating to 10% of the overall distance between the nasion and the inion (external occipital protuberance) (*Homan, Herman & Purdy, 1987*; *Chiarelli et al., 2015b*).

The two detectors were three cm distant from the *Fp* point, one on the left and one on the right side of the prefrontal cortex. Participants were instructed to keep their head as still as possible during exercise, in order to minimize the possible motion artefacts of the cerebral NIRS signal.

Before the initiation of the data recording, the near infrared spectroscopy system was calibrated with a factory-manufactured calibration optical phantom with known optical properties (absorption and scattering coefficients) according to the procedure proposed by *Hueber et al. (1999)*, which was implemented in the software of the ISS Imagent system. Absolute values of oxyhemoglobin, (HbO) deoxyhemoglobin (Hb) and total hemoglobin (HbT) concentration in the muscle and in the prefrontal cortex

were monitored continuously through the testing procedure. The $StO_2$ was calculated as (HbO)/(HbT) and was expressed as percentage.

HbO, Hb, HbT and $StO_2$ signals were low-pass filtered (3rd order Butterworth filter, cutoff frequency: 0.14 Hz) (*Bandrivskyy et al., 2004*; *Kvandal et al., 2006*) in order to eliminate some systemic noises. Then, we used a Savitzky-Golay Filter (4th order, frame size: 121) (*Ferrante et al., 2009*) to smooth the signals, thus eliminating the fluctuations of the signals due to the movement (*Chiarelli et al., 2015a*).

To assess exercise-associated oxygenation changes in the muscle, as well as changes in the left and right prefrontal cortices, we calculated $\Delta$HbO, $\Delta$Hb, $\Delta$HbT and $\Delta StO_2$ during execution of the exercise. These parameters were calculated as the difference between the absolute maximum variations in hemoglobin from the baseline. Absolute maximum variations were calculated as the mean values of 5 s around the peak value ($\pm 2.5$ s), whereas the baseline was calculated as the mean value from the last 5 s to 1 s before the onset of each exercise set.

## Statistical analysis

Data were expressed as mean $\pm$ SD. The normality of the distribution of the data was checked by the Shapiro–Wilk's normality test. All the data met the assumption of normality.

A two-way (i.e., set and condition) analysis of variance (ANOVA) with repeated measures on two factors was used to investigate the effect of conditions throughout sets in the number of repetitions and in RPE.

To test whether the amount of muscle oxygenation change in set1, set2 and set3 was influenced by the speed of movement, we used two-way (i.e., set and condition) ANOVA with repeated measures on two factors. To test whether the amount of cerebral oxygenation change in the contralateral and ipsilateral prefrontal lobe was influenced by the speed of movement, we used a two-way (i.e., lobe and condition) ANOVA with repeated measures on two factors for each set. As a measure of effect size for ANOVA, partial eta squared ($\eta_{p^2}$) was reported.

Least significant difference (LSD) post-hoc analyses were used to compare pairs of means. Statistical analysis was performed using Graphpad Prism software (version 7.0, Graphpad, San Diego, CA, USA). A *p*-value lower than 0.05 was considered statistically significant.

## RESULTS

### Exercise parameters

The effect of condition (1 s, 3 s and 5 s exercise) on number of repetitions throughout the three sets is represented in Fig. 2A, together with pairwise comparisons between conditions provided by LSD.

The two-way ANOVA repeated measures revealed a significant interaction (set $\times$ condition) for the number of repetitions ($F_{4,28} = 3.74$, $p = 0.014$, $\eta_{p^2} = 0.29$). This implies that exercising with different speed of execution (condition) until fatigue induces different number of repetitions across the sets. Furthermore, the main effect of set ($F_{2,14} = 13.53$,

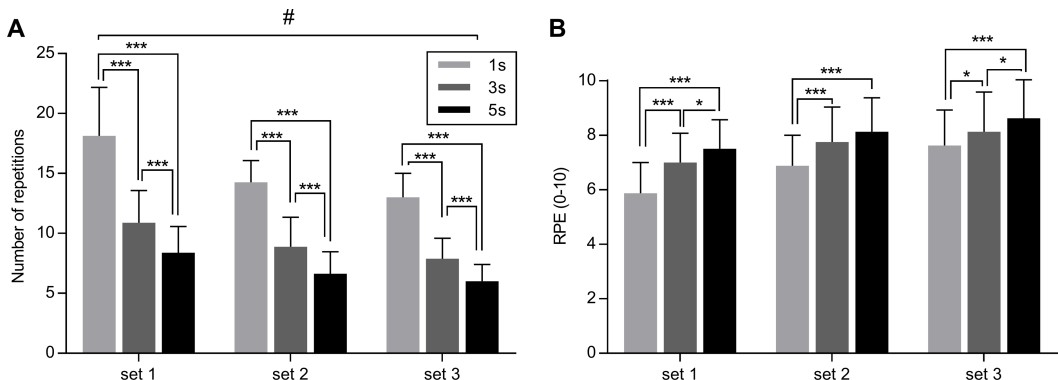

**Figure 2 Effect of different speeds of movement on maximal number of repetitions (A) and on RPE (B) during three sets of knee extension exercise.** Values are mean ± SD. Significant pairwise comparisons between conditions are shown. #$p < 0.05$ for interaction set × condition. *$p < 0.05$, ***$p < 0.001$ for pairwise comparisons between conditions. 1 s, 1 s concentric and 1 s eccentric phase. 3 s, 3 s concentric and 3 s eccentric phase. 5 s, 5 s concentric and 5 s eccentric phase.

$p < 0.001$, $\eta_{p^2} = 0.44$) and the main effect of condition were significant ($F_{2,14} = 85.53$, $p < 0.001$, $\eta_{p^2} = 0.71$).

No significant interaction (set × condition) was found for RPE ($F_{4,28} = 1.615$, $p = 0.198$, $\eta_{p^2} = 0.09$), whereas the main effect of set ($F_{2,14} = 45.71$, $p < 0.0001$, $\eta_{p^2} = 0.63$) and condition ($F_{2,14} = 8.48$, $p = 0.004$, $\eta_{p^2} = 0.40$) were found to be significant. In general, RPE was higher in the 5 s exercise, with respect to 3 s and 1 s exercise, increasing from set1 to set3. The effect of condition (1 s, 3 s and 5 s exercise) on RPE across the three sets is represented in Fig. 2B, together with pairwise comparisons between conditions provided by LSD.

## Muscle oxygenation

The acute effects of knee extension exercise on muscle hemodynamic responses (Hb and HbO) in a representative subject is illustrated in Fig. 3A. Overall, the time courses were similar in all the three exercise conditions (1 s, 3 s and 5 s exercise) for that particular variable, and are therefore described only in general terms. At the beginning of the exercise (in each set), Hb increased very rapidly and tended to level off or decrease slightly towards the end of each set. At the cessation of the exercise, during recovery, Hb began to decrease rapidly until reaching the baseline value. In parallel with Hb, HbO decreased rapidly from the initiation of the exercise, levelling off at the end of each exercise set and began to increase in the recovery returning to the baseline.

The amount of Hb and HbO change, which was quantified by ΔHb and ΔHbO, were different between conditions across sets. In fact, the two-way ANOVA repeated measures revealed a significant interaction (set × condition) for ΔHb ($F_{4,28} = 4.036$, $p = 0.01$, $\eta_{p^2} = 0.30$), implying that exercising with different speed of movement (condition) until fatigue induces different amount of deoxygenation. The main effect of set was significant ($F_{2,14} = 10.14$, $p = 0.001$, $\eta_{p^2} = 0.44$), whereas the main effect of condition was not significant ($F_{2,14} = 3.18$, $p = 0.07$, $\eta_{p^2} = 0.13$) (Fig. 4A).

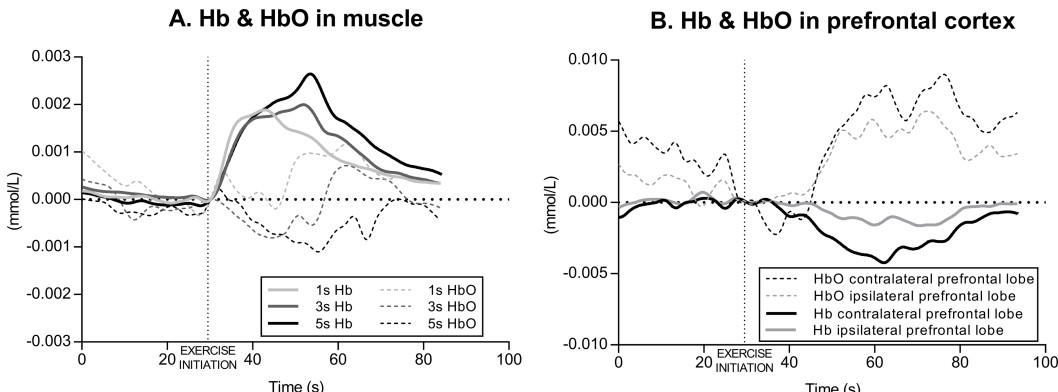

**Figure 3** **Example of muscle and cerebral oxygenation during exercise.** (A) Muscle hemodynamic responses for Hb (deoxyhemoglobin) and HbO (oxyhemoglobin) during the set2 of each condition (1 s, 3 s and 5 s exercise) of knee extension exercise in a representative subject. (B) Prefrontal cortex hemodynamic responses for Hb and HbO during the set2 of the 3 s exercise of knee extension in a representative subject.

No significant interaction (set $\times$ condition) was found for $\Delta$HbO ($F_{4,28} = 0.667$, $p = 0.619$, $\eta_{p^2} = 0.03$), but there was a significant main effect of set ($F_{2,14} = 7.404$, $p = 0.006$, $\eta_{p^2} = 0.39$) and condition ($F_{2,14} = 6.995$, $p = 0.007$, $\eta_{p^2} = 0.38$). Specifically, $\Delta$HbO was higher in 1 s set1 than 1 s set3 ($p = 0.01$), and it was higher in 5 s set1 > 5 s set3 ($p = 0.02$). Concerning the main effect of condition, $\Delta$HbO in 1 s was lower than 5 s in set1 ($p = 0.006$), and than 3 s and 5 s in set2 and set3 ($p < 0.05$) (Fig. 4B).

For $\Delta$HbT, no significant interaction (set $\times$ condition) was found for $\Delta$HbT ($F_{4,28} = 2.09$, $p = 0.108$, $\eta_{p^2} = 0.08$), as well as no significant main effect of set ($F_{2,14} = 0.253$, $p = 0.779$, $\eta_{p^2} = 0.006$) and condition ($F_{2,14} = 0.109$, $p = 0.897$, $\eta_{p^2} = 0.001$) (Fig. 4C).

No significant interaction (set $\times$ condition) was found for $\Delta$StO$_2$ ($F_{4,28} = 0.281$, $p = 0.887$, $\eta_{p^2} = 0.001$), whereas the main effect of set ($F_{2,14} = 12.58$, $p < 0.001$, $\eta_{p^2} = 0.45$) and condition ($F_{2,14} = 5.63$, $p = 0.016$, $\eta_{p^2} = 0.31$) were found to be significant. Specifically, $\Delta$StO$_2$ was higher in 1 s set1 than 1 s set2 ($p = 0.003$) and 1 s set3 ($p = 0.002$); 3 s set1 was higher than 3 s set3 ($p = 0.013$); 5 s set 1 was higher than 5 s set2 ($p = 0.007$) and 5 s set3 ($p = 0.003$). With regards to the main effect of condition, $\Delta$StO$_2$ was lower in 1 s than 3 s and 5 s in set1, set2 and set3 ($p < 0.05$) (Fig. 4D).

## Cerebral oxygenation

The acute effects of knee extension exercise on contralateral and ipsilateral prefrontal cortex hemodynamic responses (Hb and HbO) in a representative subject is illustrated in Fig. 3B. Overall, as in muscle, the time courses were similar in all three exercise conditions (1 s, 3 s and 5 s exercise), in both the left and right prefrontal cortex, and in each set, for that particular variable, and are therefore described only in general terms. At the beginning of the exercise (in each set), Hb decreased slightly and levelled off towards the end of each set, whereas HbO increased during each exercise set.

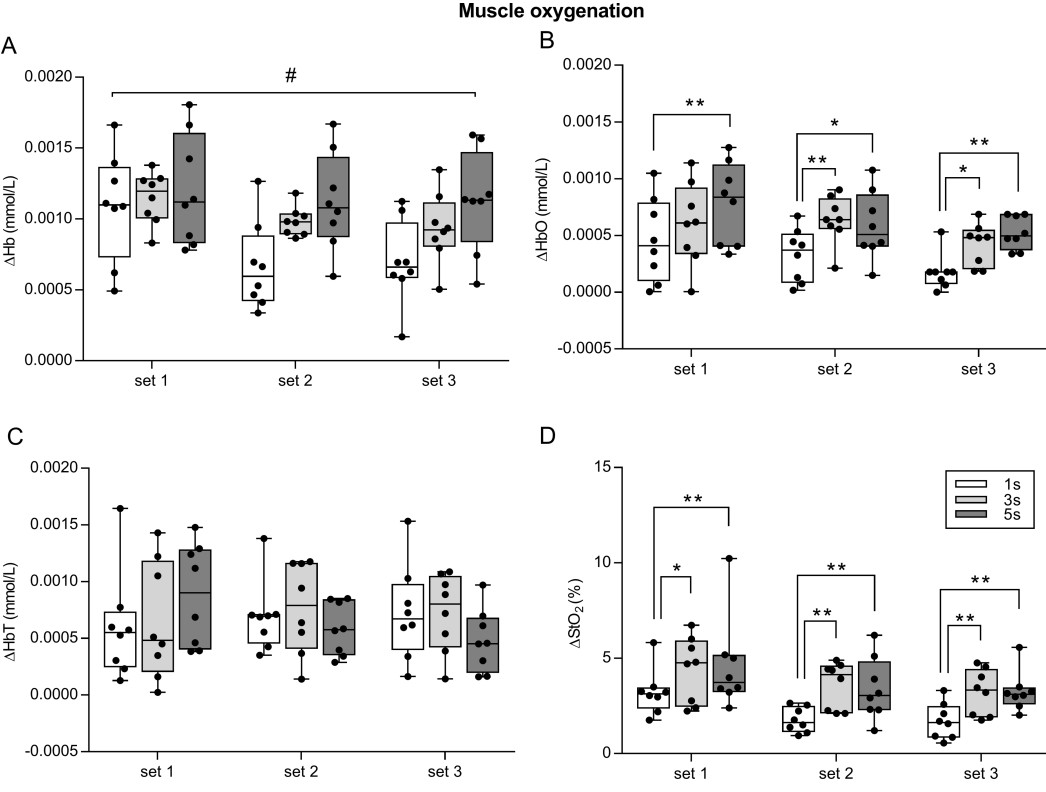

**Muscle oxygenation**

**Figure 4  Effect of different speeds of movement on muscle oxygenation change during three sets of knee extension exercise.** $\Delta Hb$, deoxyhemoglobin change (A); $\Delta HbO$, oxyhemoglobin change (B); $\Delta HbT$, total hemoglobin change (C); $\Delta StO_2$, oxygen saturation change (D). The box shows the median and interquartile range, with the whiskers indicating the range of values. Dots are values of each subject. When significant main effect of condition was found, significant pairwise comparisons between conditions are shown. #$p < 0.05$ for interaction set $\times$ condition. *$p < 0.05$, **$p < 0.01$ for pairwise comparisons between conditions. 1 s, 1 s concentric and 1 s eccentric phase. 3 s, 3 s concentric and 3 s eccentric phase. 5 s, 5 s concentric and 5 s eccentric phase.

A two-way ANOVA repeated measures (lobe $\times$ condition) was performed in each of the three sets to investigate the effects of lobe and condition for the oxygenation parameters ($\Delta Hb$, $\Delta HbO$, $\Delta HbT$, $\Delta StO_2$), and their results are reported in Table 1. Mean values $\pm$ SD of $\Delta Hb$, $\Delta HbO$, $\Delta HbT$, $\Delta StO_2$ in each condition and in each set in the two prefrontal lobes are illustrated in Fig. 5, together with pairwise comparisons.

## DISCUSSION

The main findings of this study were as follows: (1) exercising with slow speed of movement induces higher muscle deoxygenation than exercising with normal speed of movement; (2) exercise-associated oxygenation changes were higher in the contralateral compared to the ipsilateral prefrontal cortex.

**Table 1  F-values, partial eta squared ($\eta_{P^2}$) and observed power of Two-way ANOVA repeated measures (Lobe × Condition) for each set.** $\Delta$Hb (deoxyhemoglobin change), $\Delta$HbO (oxyhemoglobin change), $\Delta$HbT (total hemoglobin change), and $\Delta$StO$_2$ (oxygen saturation change).

| | Lobe | | | Condition | | | Lobe × Condition | | |
|---|---|---|---|---|---|---|---|---|---|
| | $F_{(1,7)}$ | $\eta_{P^2}$ | Observed power (1-$\beta$ err prob) | $F_{(2,14)}$ | $\eta_{P^2}$ | Observed power (1-$\beta$ err prob) | $F_{(2,14)}$ | $\eta_{P^2}$ | Observed power (1-$\beta$ err prob) |
| Set1 | | | | | | | | | |
| $\Delta$Hb | 6.907[*] | 0.21 | 0.51 | 5.955[*] | 0.29 | 0.83 | 4.174[*] | 0.19 | 0.59 |
| $\Delta$HbO | 7.821[*] | 0.24 | 0.72 | 5.263[*] | 0.28 | 0.81 | 2.808 | 0.11 | 0.34 |
| $\Delta$HbT | 6.802[*] | 0.21 | 0.51 | 5.426[*] | 0.28 | 0.81 | 3.567 | 0.17 | 0.52 |
| $\Delta$StO$_2$ | 6.913[*] | 0.20 | 0.61 | 9.977[**] | 0.42 | 0.98 | 2.117 | 0.09 | 0.28 |
| Set2 | | | | | | | | | |
| $\Delta$Hb | 5.53[*] | 0.19 | 0.59 | 0.853 | 0.02 | 0.09 | 1.811 | 0.09 | 0.28 |
| $\Delta$HbO | 8.534[*] | 0.22 | 0.67 | 2.285 | 0.08 | 0.24 | 0.649 | 0.02 | 0.09 |
| $\Delta$HbT | 19.2[**] | 0.39 | 0.95 | 1.359 | 0.06 | 0.19 | 1.81 | 0.09 | 0.28 |
| $\Delta$StO$_2$ | 9.17[*] | 0.28 | 0.81 | 1.59 | 0.06 | 0.19 | 0.139 | 0.001 | 0.05 |
| Set3 | | | | | | | | | |
| $\Delta$Hb | 10.55[*] | 0.29 | 0.83 | 1.638 | 0.06 | 0.19 | 0.706 | 0.02 | 0.09 |
| $\Delta$HbO | 7.449[*] | 0.24 | 0.72 | 1.158 | 0.04 | 0.14 | 0.004 | 0.001 | 0.05 |
| $\Delta$HbT | 2.493 | 0.09 | 0.28 | 3.71 | 0.17 | 0.53 | 0.355 | 0.006 | 0.06 |
| $\Delta$StO$_2$ | 5.095[*] | 0.19 | 0.59 | 3.6 | 0.17 | 0.53 | 0.323 | 0.006 | 0.06 |

**Notes.**
[*] $p < 0.05$.
[**] $p < 0.01$.

## Muscle oxygenation

Consistent with the study hypothesis, exercising with different speeds of movement (1 s, 3 s and 5 s exercise: condition) until fatigue induced different amount of change in muscle oxygenation. In fact, differences between conditions were found in each of the three sets for $\Delta$HbO and $\Delta$StO$_2$. A significant (condition × set) interaction was also found for $\Delta$Hb.

In the present study we found that exercising with slow speed of movement induced larger muscle deoxygenation compared to exercising with normal speed of movement (i.e., 1 s exercise). Many lines of evidence point to such a conclusion. The first observation is that the amount of change in oxygenation was larger in 5 s than 3 s exercise and in 3 s than 1 s exercise (Fig. 3). Accordingly, the results of the two-way ANOVA for repeated measures revealed a significant interaction between condition and set for $\Delta$Hb and a main effect of condition for $\Delta$HbO, thus indicating that speed of movement has an effect on the amount of oxygenation change (Fig. 4).

Our results are in line with the study by *Tanimoto & Ishii (2006)*, which investigated the acute oxygenation changes in the vastus lateralis during a knee extension exercise with different speeds of execution (different repetition durations). It was found that exercising with slow speed of movement (3 s for concentric and eccentric phase) with relatively low intensity (∼50% of 1RM) elicited significantly larger muscle deoxygenation than the other two protocols, which were performed with normal speed of movement (1 s for each phase) and different intensities (∼50% of 1RM, and ∼80% of 1RM). In that study, a 12-week period of resistance exercise with slow speed of movement was also accompanied by positive

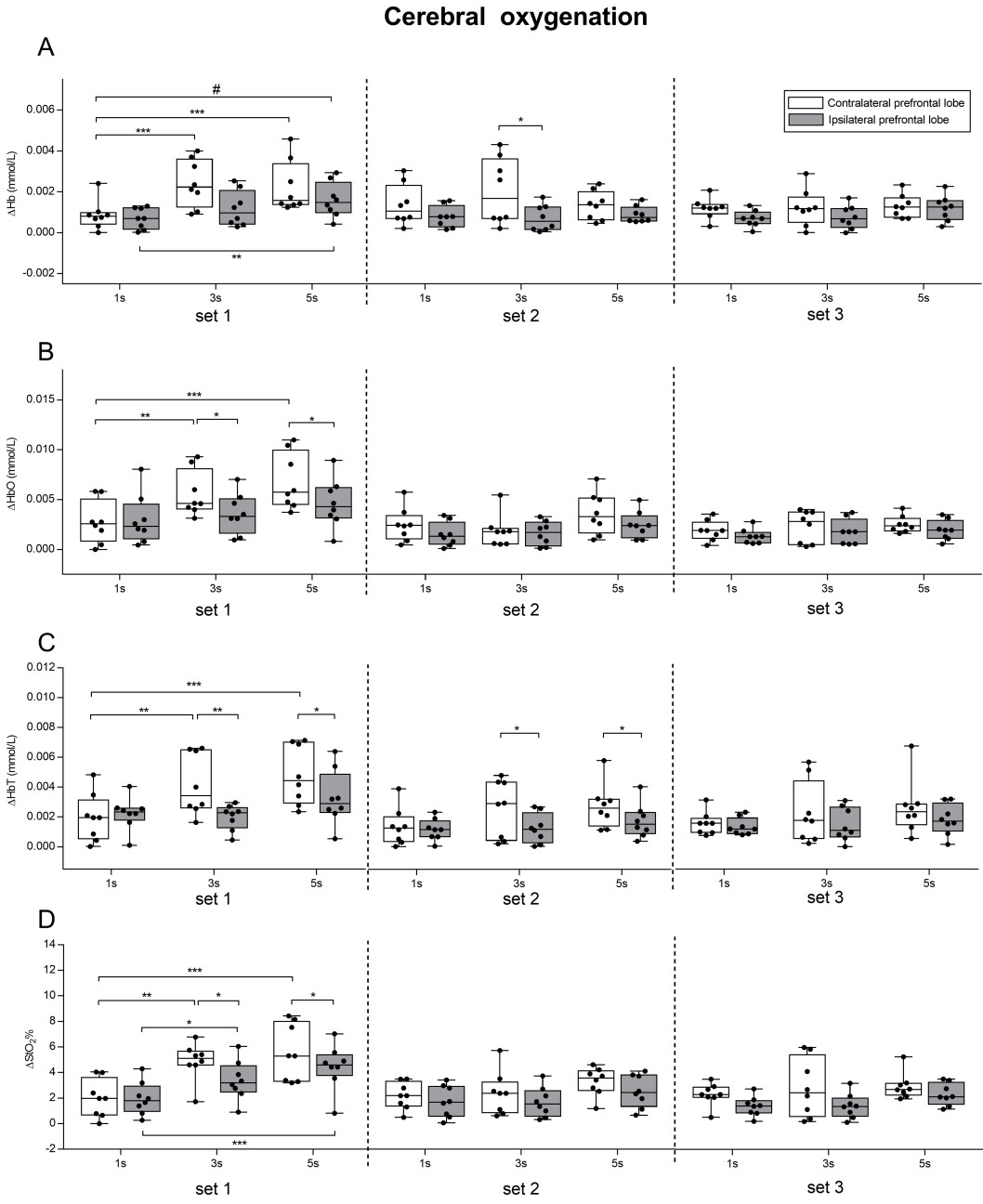

**Cerebral oxygenation**

**Figure 5** **Effect of different speeds of movement on cerebral oxygenation change in left and right prefrontal lobe during three sets of knee extension exercise.** $\Delta$Hb, deoxyhemoglobin change (A); $\Delta$ HbO, oxyhemoglobin change (B); $\Delta$HbT, total hemoglobin change (C); $\Delta$StO$_2$, oxygen saturation change (D). The box shows the median and interquartile range, with the whiskers indicating the range of values. Dots are values of each subject. Significant pairwise comparisons between lobes and between conditions are shown. #$p < 0.05$ for interaction lobe × condition. *$p < 0.05$, **$p < 0.01$, ***$p < 0.001$. 1 s, 1 s concentric and 1 s eccentric phase. 3 s, 3 s concentric and 3 s eccentric phase. 5 s, 5 s concentric and 5 s eccentric phase.

effects on muscular size and strength, which were similar to those obtained by exercise with normal speed. Based on these findings, the authors suggested that muscle hypertrophy may be mediated by the hypoxic condition of the intramuscular environment, increasing metabolic stress and stimulating systemic hormone secretion, cell swelling, production of ROS and increased recruitment of fast-twitch fibers (*Pearson & Hussain, 2015*).

In addition, to investigate muscle oxygenation change in exercise with a slow speed of movement (3 s for each phase) compared to an exercise with a normal speed movement (1 s for each phase)—already addressed by *Tanimoto & Ishii (2006)*—we also investigated oxygenation change associated to an exercise with an even slower speed of movement (i.e., 5 s for each phase). We found that the 5 s exercise was associated with larger muscle deoxygenation compared to the 1 s exercise, as shown by the significant interaction for $\Delta Hb$, and the significant main effect of condition for $\Delta HbO$, and $\Delta StO_2$ (Fig. 4). Specifically, significant differences between conditions were only found between 1 s and 3 s exercise (for set2 and set3), and between 1 s and 5 s exercise (for each set). No significant differences were found between 3 s and 5 s exercise for all of the parameters, which seems to indicate that exercising with an even slower speed of movement than 3 s for each phase did not elicit an even lower muscle oxygenation (Fig. 4). Moreover, 5 s exercise was associated with a lower number of repetitions and higher RPE in each set, with respect to both 1 s and 3 s exercise (Fig. 2).

It is important to notice that no differences between conditions were found for $\Delta HbT$, which can be considered an indirect measure of change in blood flow (*Felici et al., 2009*). Given that $\Delta HbT$ was similar among conditions, we speculate that the blood flow modifications induced by the exercise were similar among conditions. Therefore, exercising with slow movement caused greater deoxygenation in muscles, which was probably due to larger oxygen consumption, while the changes in blood flow remained the same as during exercise with normal speed of movement.

The significant main effect of set found for $\Delta HbO$ and $\Delta StO_2$ demonstrated a significant decrement of oxygenation change from set1 to set3. This can be explained by exercise-related fatigue throughout the whole exercise session. The decrement of changes in oxygenation from set1 to set3 was also consistent with the exercise parameters (number of repetitions and RPE) which changed across the training session. In particular, the number of repetitions were higher in set1, decreasing in set2 and in set3, and accordingly, RPE increased from set1 to set3. These findings are consistent with the data presented in a recent study (*Ganesan et al., 2015*). In that study, the authors investigated the oxygenation changes in the vastus medialis muscle and the prefrontal cortex of the brain during three sets of knee extension exercise, with and without blood flow restriction induced by an inflatable cuff. Although no between sets comparisons were shown, the number of repetitions decreased from set1 to set3, and accordingly RPE increased from set1 to set3.

Our results demonstrated that exercising with a slow speed of movement induced higher amount of muscle deoxygenation than normal speed of movement (1 s for each phase, i.e., 2 s for each repetition). It is worth noticing that, despite the discrepancy between the two experimental conditions (i.e., slow speed of movement in our study vs. blood flow restriction with an inflatable cuff in the study by *Ganesan et al. (2015)*),

these two protocols elicited a significant decrease in muscle oxygenation. It has been suggested that mechanical tension and metabolic stress are the two primary mechanisms for stimulating muscle hypertrophy, the latter magnified under conditions of lack of oxygen within an intramuscular environment (*Pearson & Hussain, 2015*). Metabolic stress mediates muscle growth via a series of secondary mechanisms, such as increased hormone productions (*Takarada et al., 2000*), increased fast-twitch fibers recruitment (*Moritani et al., 1992*), cell swelling (*Loenneke et al., 2012*) and increased ROS production (*Kawada & Ishii, 2005*). Although resistance training with slow speed of movement is characterized by low intensity, as that employed in the present study, the higher deoxygenation reached by such condition may be considered an additional stimulus for inducing high metabolic stress, thus maximizing one of the two primary mechanisms by which hypertrophy is stimulated. Therefore, the manipulation of speed of movement, and the related repetition duration, should be considered when planning resistance training to maximize metabolic stress within intramuscular environment.

## Cerebral oxygenation

We found that exercising with slow speed of movement (5 s exercise and 3 s exercise) induced larger changes in both the left and right prefrontal cortex oxygenation, compared to exercising with normal speed of movement (1 s exercise). Such changes were more pronounced in the contralateral prefrontal lobe. The fact that exercising with slow speed of movement elicited greater oxygenation change in both the contralateral and ipsilateral prefrontal cortices, confirmed that fatigue seems to be mediated not only peripherally, but also centrally (*Ganesan et al., 2015*). This is particularly evident for the set1, in which a significant interaction (lobe $\times$ condition) was found for $\Delta Hb$. Moreover, in the contralateral prefrontal cortex, the 1 s exercise resulted in lower oxygenation change than 3 s exercise and 5 s exercise ($\Delta HbO$, $\Delta StO_2$, $\Delta HbT$). It is worth pointing out that, despite the significant main effect of lobe, LSD pairwise comparisons revealed significant differences between lobes only in the 3 s and 5 s exercise conditions ($\Delta HbO$, $\Delta StO_2$, $\Delta HbT$) (Fig. 5). Conversely, in set2 and set3, no significant main effects of conditions were found, whereas there were significant main effects of lobe (Table 1). The fact that in set2 and in set3 no differences between conditions were found for all the parameters may be explained by physiological mechanisms related to change in blood flow, which are still very poorly investigated. In fact, studies pertaining to the effect of skin blood flow on cerebral oxygenation during exercise focused only on a walking treadmill task (*Kohno et al., 2007*; *Gomes, Matsuura & Bhambhani, 2012*).

An interesting study has recently investigated the acute effect of blood flow restriction (created artificially using an inflatable cuff) during exercise, not only on muscle oxygenation, but also on brain prefrontal cortex oxygenation (*Ganesan et al., 2015*).

Similarly to that study (*Ganesan et al., 2015*), our data suggests that the pronounced decrement in muscle oxygenation can modulate exercise-induced changes in prefrontal cortex oxygenation, which was proposed to be related to the perception of fatigue (RPE) (*Pereira, Gomes & Bhambhani, 2009*).

The results of the present study support the complementary interaction between the ipsilateral and contralateral cortex during exercise to fatigue. In fact, we found that the increase of oxyhemoglobin in the contralateral lobe (and the consequent decrease of deoxyhemoglobin) was accompanied by a smaller but consistent increase in the ipsilateral lobe. We also found that the change in oxygenation was higher in the contralateral compared to the ipsilateral prefrontal lobe in each of the three sets (Fig. 5 and significant main effect of lobe shown in Table 1). The fact that oxygenation increased in both prefrontal cortices, albeit the increase was lower for the ipsilateral than the contralateral, is in line with the findings of a previous study (*Kuboyama & Shibuya, 2015*), which revealed an increase in oxyhemoglobin accompanied by a small decrease in deoxyhemoglobin in both hemispheres during prolonged fatiguing repetitive handgrip exercise performed at maximal voluntary contraction (*Kuboyama & Shibuya, 2015*). However, despite changes in both hemispheres, there was higher delayed oxygenation in the ipsilateral compared to the contralateral cortex, thus suggesting that unilateral resistance exercise has a stronger association with the contralateral rather than the ipsilateral lobe (*Kuboyama & Shibuya, 2015*). Our data provided evidence of a complementary interaction between the ipsilateral and contralateral cortex during fatiguing exercise (*Benwell, Mastaglia & Thickbroom, 2006*; *Shibuya & Kuboyama, 2007*), as well as a higher involvement of the contralateral prefrontal cortex compared to the ipsilateral.

### Limitations and further directions of research

It is possible that the variability of some oxygenation parameters in muscle, but especially in the prefrontal cortex, among subjects may have hampered our ability to detect differences between conditions in the second and third set of exercise. Although our sample size was found to be sufficient, further studies will have to consider such variability by increasing the sample size to achieve more statistical power. Furthermore, despite a previous study showing that there were no differences between males and females in muscle and cerebral oxygenation during resistance exercise (*Gomes, Matsuura & Bhambhani, 2012*), we put in evidence that our findings cannot be surely extended also to male subjects. Thus, future studies will be addressed to compare exercise-associated oxygenation changes of muscle and prefrontal cortex in male and female subjects, as well as in subjects with different training levels.

However, it is important to notice that this study has been conducted using a frequency-domain NIRS instrument that allowed us to carry out accurate measurements of absolute concentrations of oxyhemoglobin and deoxyhemoglobin, thus also quantifying total hemoglobin and saturation. This was useful in order to obtain a detailed portrait of hemodynamic changes in both muscle and the prefrontal cortex related to resistance exercise with different speeds of movement. In addition, a comparison between oxygenation changes in the contralateral and ipsilateral prefrontal cortex was performed.

Finally, we suggest that further investigations should consider a combination of different techniques to further explore the underlying physiological mechanisms of resistance exercise with different speeds of movement, such as muscle activity by electromyography and tissue perfusion by infrared thermography (*Formenti et al., 2016*). We also recommend

the use of a full-head fNIRS helmet to elucidate the response of different brain regions to resistance exercise performed to fatigue.

## CONCLUSIONS

In summary, the findings of the present study indicate that exercising with slow speed of movement induces higher muscle deoxygenation than exercising with normal speed of movement. Specifically, the exercise with a repetition duration lasting 10 s (5 s eccentric and 5 s concentric phase) was associated with larger muscle deoxygenation compared to the exercise with faster movement (3 s and 1 s for each phase). Since the lack of oxygen in muscle can maximize the levels of metabolic stress, which has a primary role in stimulating hypertrophy, manipulating the speed of movement, and the related repetition duration, may be useful when planning resistance-training programs to increase the metabolic stress associated to exercise.

Furthermore, we found a consistent oxygenation increase in both the right and left prefrontal lobes, and that oxygenation changes were higher when exercising with slow movement. In addition, higher changes in oxygenation were found in the contralateral rather than the ipsilateral prefrontal cortex. These results provide further evidence of the complementary interactions between the ipsilateral and contralateral prefrontal cortex during fatiguing resistance exercise, and that the speed of movement has an impact on prefrontal cortex oxygenation.

## ACKNOWLEDGEMENTS

Special thanks to Ben Oliver for proof-reading the manuscript and Antonello Chiarelli, PhD, for his technical support.

### Funding

The authors received no funding for this work.

### Competing Interests

The authors declare there are no competing interests.

### Author Contributions

- Damiano Formenti and David Perpetuini conceived and designed the experiments, performed the experiments, analyzed the data, contributed reagents/materials/analysis tools, prepared figures and/or tables, authored or reviewed drafts of the paper, approved the final draft.
- Pierpaolo Iodice conceived and designed the experiments, performed the experiments, contributed reagents/materials/analysis tools, prepared figures and/or tables, authored or reviewed drafts of the paper, approved the final draft.

- Daniela Cardone performed the experiments, analyzed the data, contributed reagents/materials/analysis tools, prepared figures and/or tables, authored or reviewed drafts of the paper, approved the final draft.
- Giovanni Michielon performed the experiments, analyzed the data, contributed reagents/materials/analysis tools, authored or reviewed drafts of the paper, approved the final draft.
- Raffaele Scurati analyzed the data, contributed reagents/materials/analysis tools, authored or reviewed drafts of the paper, approved the final draft.
- Giampietro Alberti conceived and designed the experiments, contributed reagents/-materials/analysis tools, authored or reviewed drafts of the paper, approved the final draft.
- Arcangelo Merla conceived and designed the experiments, analyzed the data, contributed reagents/materials/analysis tools, authored or reviewed drafts of the paper, approved the final draft.

## Human Ethics

The following information was supplied relating to ethical approvals (i.e., approving body and any reference numbers):

The study was approved by the Ethical Committee of the Università degli Studi di Milano (approval number: 2/12).

## Data Availability

The raw data are provided in Supplemental File.

## Supplemental Information

Supplemental information for this article can be found online at http://dx.doi.org/10.7717/peerj.5704#supplemental-information.

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
