# Peer review of "Effects of knee extension with different speeds of movement on muscle and cerebral oxygenation"

_PeerJ, doi:10.7717/peerj.5704_

## Round 0.1 · original submission · Major Revisions

Thank you for submitting this interesting manuscript and I would like to apologize for the delay in submitting the decision (I was waiting on a reviewer who ultimately did not submit their promised review).

Upon resubmission, please carefully address each of the reviewers' comments and suggestions. In particular, please ensure your writing is clear, concise, and follows normal conventions for the usage of the English language. Please also clearly state your methods and provide more robust support for their discussion material. Also note that reviewer 2 has submitted an annotated manuscript with specific comments/changes that you should address.

Reviewer 1 ·

Basic reporting

L. 57 – You stage the introduction in the manner that suggests longer repetition durations should result in greater hypertrophy, but then mention that repetition durations of 0.5 to 8 s to failure result in similar hypertrophic outcomes. This detracts from your entire purpose of this study. Do you have any theories why the study by Schoenfeld et al. (2015) observed no differences in hypertrophy across ranging repetition durations? This will to support your case for using longer repetition durations for hypertrophic changes.

L. 64 – Define “normal speed”, and “relatively low intensity”.

L. 65 – You have written that lowered peripheral oxygenation is a mechanism for muscle hypertrophy, but fail to elaborate on the mechanism itself. Please consider including more background on the actual mechanism involved here.

L. 68 to 70 – Consider the wording of this sentence for readability. It currently doesn’t flow well with words like “hence, likely, even more, thus” in the same sentence.

L. 71 – Typo: “…elicits early recruited…”

L. 72 – Does fiber type (I, IIa, IIb) play a role in the oxygenation response during resistance training?

L. 73 – Fatigue or failure?

L. 73 to 74 – It would help the reader interpret the findings in the prefrontal cortex if you provide a direction and magnitude of changes in oxygenation. In its current state, you only mention muscle oxygenation is related to changes in the prefrontal cortex.

L. 74 to 75 – What role does cerebral oxygenation play in the ability to perform maximal exercise? If anything, the current body of literature would indicate that as exercise intensity reaches maximal levels, cerebral blood flow decreases. Is this the same for oxygenation? Keep in mind that the prefrontal cortex has no role in motor control.

L. 80 to 86 – Consider removing joining words to improve paragraph readability. You have four sentences in a row starting with 1.) In fact, 2.) However, 3.) Furthermore, and 4.) Thus.

L. 82 to 84 – Please elaborate on the theory that muscle oxygenation is controlled centrally (I assume you mean from the brain).

L. 83 – Consider rewording: reduced increase.

L. 86 – Typo: missing full stop “…and blood flow (Fernandes et al., 2016) Thus…”

L. 95 to 96 – I would suggest that you are observing a response, not investigating a mechanism.

L. 180 – Typo: “…tissue thickness resulted smaller…”

L. 302 to 303 – Typo: “…In fact, difference between conditions…”

L. 320 to 322 – What are the theories for this mechanism?

L. 596 – No journal title.

L. 473, 493, 500, 509, 512, 518, 553, 557 – Find consistency with the naming of Journals (i.e. MSSE and JSCR mainly).

Experimental design

L. 146 – You have stated that 30 s of rest between RM attempts was provided to participants. Do you think this is sufficient to recovery maximal muscle force? Typical RM guidelines would lean towards 120 s to 240 s between RM attempts.

L. 201 to 204 – Based upon my own research/personal experience, performing exercise to fatigue typically causes someone to grimace, look down, look up etc. as a coping strategy. Since cerebral NIRS devices are extremely sensitive to total fluid shifts, it is important for head position to remain stable. Were participants given instruction as to their head placement during exercise?

L. 348 – Your RPE scores are not consistent with maximal performance. I would wager that a true RPE score following repetitions to concentric failure would be 10, irrespective of the repetition duration. How were the participants asked to rate RPE? Was this standardised across participants? Were participants provided any anchoring of RPE scores?

Validity of the findings

Table 1 – A table should technically be able to stand on its own. Consider listing the abbreviations in the caption (i.e. ΔHb = haemoglobin change).

Figure 1 – I assume the 10 min rest prior to the knee extension was used to collect baseline NIRS data to normalise from. If so, it’s possible that the warm-up might have influenced these values. Was there any reason why you didn’t elect to perform the warm-up after the baseline collection?

Figure 2B – Your Y axis is marked as 0 to 12, however, your RPE scale is listed as 0 to 10 in brackets. In addition, this is also inconsistent with your methodology section, where your RPE values are listed as being between 1 and 10 (L. 164).

Figure 4 and Figure 5 – Consider using graphing techniques outlined by Weissgerber et al. (2015, Beyond Bar and Line Graphs: Time for a New Data Presentation Paradigm). This will allow readers to examine the data for spread, outliers etc.

Additional comments

The authors have investigated the influence of repetition duration on oxygenation of dominant leg vastus lateralis muscle and prefrontal cortex via near-infrared spectroscopy in untrained females. To the best of my knowledge, this work is novel and adds to the existing body of evidence. The project is well designed and methodologically sound, with minor concerns. The authors have chosen appropriate statistical analyses. I find the article to be written to a satisfactory level; however, there are numerous typographical errors that could be quickly fixed to improve the overall quality of the article. I have also highlighted several areas that could be revised to improve readability. Please find my feedback below listed under line (L.) numbers. If the authors consider these changes, I would gladly recommend this article for publication in PeerJ.

Reviewer 2 ·

Basic reporting

The authors have presented a manuscript titled “Effects of knee extension with different speeds of movement on muscle and cerebral oxygenation.” Eight female participants performed three sets of unilateral knee extensions at 50% 1RM with various velocities referred to by the authors as the amount of time provided for the complete of both concentric and eccentric motions of the exercise. Frequency domain NIRS was used to assess deoxyhemoglobin, oxyhemoglobin, total hemoglobin, and oxygen saturation. It is not clear from the abstract what muscle was analyzed and this should be clearly stated. Both contralateral and ipsilateral prefrontal cortex (PFC) measurements were taken. Authors stated that exercise at a slower speed increased deoxygenation of the muscle and the contralateral lobe of the PFC showed greater increases in oxygenation than the ipsilateral PFC. The results presented by the authors is very intriguing yet there are several important areas that should be noted to improve the manuscript before acceptance.

The authors are suggested to consult resources available for the improvement of the English language used throughout the manuscript to better cater to the international audience of the journal. It is also suggested that a native English speaking colleague to review the manuscript before submitting. Some examples where language could be improved have been noted in the marked manuscript (Microsoft Word track changes) that has been included in this review. Further grammatical and language corrections were not made for the rest of the manuscript which should be examined thoroughly by the authors before resubmission.

Formatting should be consistent throughout the manuscript. For instance, in the abstract authors state their hypotheses as numerals (i. and ii.; suggested to be 1 and 2; page 2) but then later at the end of the introduction the same hypotheses are listed as a. and b. For the hypotheses, they are vague in terms of outcomes. The authors state they hypothesize that the changes will be higher but it should be stated clearly what that means (i.e. higher levels of oxygenation).

Experimental design

How did the authors confirm that the futsal players were not engaged in resistance training as part of their sport involvement? For the 1RM testing, what equipment was used? How was the predicted 1RM calculated? The authors state that only 30 seconds of rest was provided in between successive trials which is a very small amount of time for recovery during 1RM testing. The National Strength and Conditioning Association recommends 2 minute rest intervals after the warm-up and 2 to 4 minute rest periods for the 1RM attempts (Essentials of Strength Training and Conditioning, 4th edition, Haff and Triplett). Furthermore, research examining the rest interval length for 1RM squats states that at least a 1 minute rest interval, similar to that reported for bench press, is sufficient for recovery (Matuszak 2003). It seems at the very least that a minimum of 1 minute with participant allowance to take further rest if requested would best ensure that a true 1RM was obtained. The authors should explain their choice of 30 seconds for a rest interval and provide evidence that this is a sufficient rest period for producing valid 1RM loads. The authors should describe their general protocol for 1RM testing.

What was the general range of motion achieved for the exercise? The authors state that full knee extension, noted as extended to 180 degrees or often referred to as 0 degrees of knee flexion, but what was the starting position for the lower limb? Following 1RM, how were participants familiarized with the resistance exercise protocol?

For the exercise sessions, how was “concentric failure” defined? Were there any allowable failed attempts and if so how many? What occurred if participants were unable to keep pace with the metronome? It was listed that exhaustion occurred during concentric failure but what if participants were successfully completing the concentric portion according to the metronome but were unable to properly control the eccentric portion of the movement?

It was stated that changes in NIRS variables were calculated as the difference between the absolute maximum variations from baseline but there was no explanation of how baseline levels were obtained. Furthermore, the length of time to obtain maximum and baseline measurements was not stated.

Validity of the findings

The authors refer to the study by Tanimoto and Ishii who found that the longer duration of exercise movement resulted in increased deoxygenation but at the same time achieved levels of muscular size and strength to that of the faster movement duration. The authors further state that Tanimoto and colleagues suggest that it is the large muscle deoxygenation that has a fundamental role in stimulating muscle hypertrophy but if this is the case, would the longer duration group who experienced the greater levels of deoxygenation not also experience greater strength and hypertrophy changes? After analyzing the article, I am unable to see where those authors directly state how reaching a large muscle deoxygenation plays a fundamental role for muscle hypertrophy. The authors need to better support their discussion about the importance of deoxyhemoglobin in relation to their remarks and be cautious of their interpretation of cited work. Additionally, in the conclusion section, the authors make a strong statement about lack of muscle oxygen having a key role in stimulating hypertrophy. The authors need to better represent the current literature to make such statements. This occurs in lines 451-454 which also states, at least based on how it is written, that the current research linked lack of oxygen to hypertrophy which is not the case.

Additional comments

It should be noted that on line 363 that the study discussed did not use an elastic cuff but an inflatable cuff through the Hokanson system which is commonly used in blood flow restriction research.

Lines 185-186: This sentence is unclear as it is written and should be reworded.

Lines 190: Authors should properly define Fp point.

Lines 219 -222: Authors state the two factors (i.e. lobe and condition) twice in the same sentence for both ANOVA descriptions. Only one instance is necessary.

Asterisks are misaligned in the figures making it more of a challenge to interpret.

The raw data has much information that is not included in the manuscript and further detail is needed to fully understanding the dataset. For instance, it appears that standard jump, countermovement jump, MVC, and two isokinetic measurements were taken but none of these were discussed in the manuscript. Columns X through AA have headings 30 m (s) and 60 (s) and it is unclear what these are referring to. T0 and T1 are not defined.

Overall the authors have presented data that may improve upon our understanding of how contraction duration affects hemodynamics as measured by NIRS. Although the data is interesting, the authors need to better support their discussion about the importance of their results and improve on the usage of the English language. Furthermore, there is much lacking in the methods section for fully understanding the methods that were utilized and the authors should include greater detail in this area. These changes should be made before acceptance is granted.

Annotated reviews are not available for download in order to protect the identity of reviewers who chose to remain anonymous.

---

## Round 0.2 · Minor Revisions

Thank you for your attention to detail in your resubmission. Both reviewers and I feel that you have very nearly satisfied the requirements needed to make this manuscript acceptable. There are a few very minor suggestions made by Reviewer 2 that I would like you to address before I can accept this manuscript. Please address these further concerns and resubmit your manuscript. I look forward to the final version!
Scotty

Reviewer 1 ·

Basic reporting

No comment.

Experimental design

No comment.

Validity of the findings

No comment.

Additional comments

The authors have made significant amendments to their original manuscript, and have addressed each of my prior concerns and questions. As such, I am happy to approve this article for publication.

Reviewer 2 ·

Basic reporting

NA

Experimental design

NA

Validity of the findings

NA

Additional comments

Overall, the authors appear to have addressed the concerns of the reviewers. The attention to the concerns is greatly appreciated. Before full publication, there are still some minor areas that should be addressed regarding grammar and formatting and data reporting. The references are still not uniformly formatted (for example, some titles are all capitalized where some are not). Please thoroughly check to make sure this is correct. Furthermore, there are misspelled words found (for example, calliper). I also did not see any of the data presented on the skinfold thickness in the manuscript or the raw data. This would be helpful in the interpretation of data given the adiposity of the participants measured. Figure 5A y-axis displays "HbB" but given previous reporting in the manuscript should be "Hb". Figure 3 incorporates red and green colors which will make it challenging if not impossible for some readers that have color deficiency.

---

## Round 0.3 · accepted · Accept

Congratulations on your acceptance and well done.